# Causal links between parietal alpha activity and spatial auditory attention

**Yuqi Deng[1], Robert MG Reinhart[2], Inyong Choi[3], Barbara G Shinn-Cunningham[1,4]***

[1]Biomedical Engineering, Boston University, Boston, United States; [2]Psychological and Brain Sciences, Boston University, Boston, United States; [3]Communication Sciences and Disorders, University of Iowa, Iowa City, United States; [4]Neuroscience Institute, Carnegie Mellon University, Pittsburgh, United States

**Abstract** Both visual and auditory spatial selective attention result in lateralized alpha (8–14 Hz) oscillatory power in parietal cortex: alpha increases in the hemisphere ipsilateral to attentional focus. Brain stimulation studies suggest a causal relationship between parietal alpha and suppression of the representation of contralateral visual space. However, there is no evidence that parietal alpha controls auditory spatial attention. Here, we performed high definition transcranial alternating current stimulation (HD-tACS) on human subjects performing an auditory task in which they directed attention based on either spatial or nonspatial features. Alpha (10 Hz) but not theta (6 Hz) HD-tACS of right parietal cortex interfered with attending left but not right auditory space. Parietal stimulation had no effect for nonspatial auditory attention. Moreover, performance in post-stimulation trials returned rapidly to baseline. These results demonstrate a causal, frequency-, hemispheric-, and task-specific effect of parietal alpha brain stimulation on top-down control of auditory spatial attention.

## Introduction

### Parietal alpha activity changes with the focus of spatial attention

Parietal cortex interacts with frontal cortex to control spatial attention in both vision and audition (*Farah et al., 1989*; *Green et al., 2011*). Functional magnetic resonance imaging (fMRI) reveals a series of retinotopically mapped regions ascending along the intraparietal sulcus (IPS), which are biased towards representing contralateral exocentric space (*Sereno et al., 2001*; *Swisher et al., 2007*). While the earlier mapped regions are strongly engaged only by vision, the higher maps are recruited when participants engage spatial auditory attention (*Michalka et al., 2016*).

Alpha oscillations (8–14 Hz) are associated with a range of neural functions (*Klimesch, 2012*; *Weisz et al., 2011*). Parietal cortex generates strong alpha oscillations measurable using electro- and magneto-encephalography (EEG and MEG) (*Jensen and Mazaheri, 2010*; *Kelly et al., 2006*; *Worden et al., 2000*). When listeners focus visual attention, alpha power lateralizes, increasing in the parietal hemisphere ipsilateral to the direction of attention and decreasing contralaterally (*Kelly et al., 2006*). Auditory spatial attention also results in lateralized parietal alpha activity (*Banerjee et al., 2011*; *Mehraei et al., 2018*; *Deng et al., 2019*; *Wöstmann et al., 2016*; *Frey et al., 2014*; *Strauß et al., 2014*); indeed, alpha lateralization shifts systematically as the focus of auditory spatial attention shifts from far-left to far-right (*Mehraei et al., 2018*; *Deng et al., 2020*) (see *Figure 1A*). These results suggest that focusing spatial attention in both vision and audition leads to similar parietal alpha activity.

While auditory spatial processing relies on retinotopic regions of parietal cortex, processing non-spatial features does not, even when listeners are attending the same source in the same sound mixture (*Michalka et al., 2016*; *Hill and Miller, 2010*; *Noyce et al., 2017*). Indeed, for a target defined

*For correspondence:
bgsc@andrew.cmu.edu

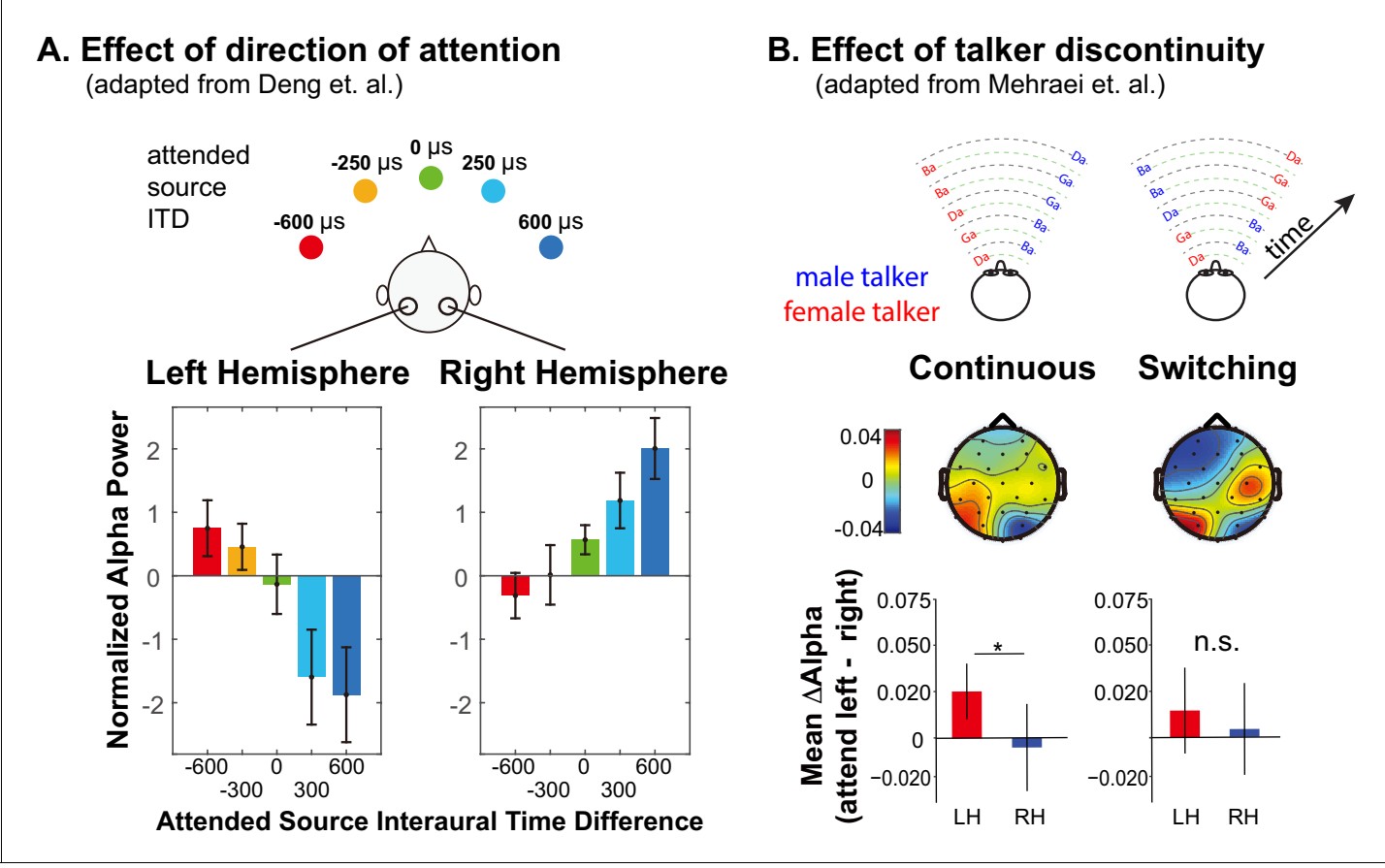

**Figure 1.** Summary of previous results exploring lateralization of alpha power during auditory spatial attention processing. (A) Adapted with permission from *Deng et al. (2020)*. Average normalized alpha activity (relative to global field power of alpha, calculated over N = 26 subjects) in left and right parietal EEG sensors measured in an auditory spatial attention task. Spatial attention was covertly directed to one of five different lateral positions (see top of the panel), controlled by changing the interaural time difference (ITD) of the target stimulus. The average of alpha power during a preparatory period (after an auditory cue indicating the target position, but prior to the start of the sound presentation) is shown separately for left and right parietal sensors in the left and right bottom panels, respectively. (B) Adapted with permission from *Mehraei et al. (2018)*. Alpha lateralization goes away when a discontinuity in the target talker disrupts spatial auditory attention. Example stimuli are shown at the top for trials in which the nonspatial (talker) features of two competing speech streams are continuous (left) and when they switch, which, if it occurred, always was after the first two target syllables (right). Topographies show the normalized difference in alpha power when listeners focus attention to the left minus when they focus attention to the right (relative to the sum of the alpha power for attend-left plus attend-right), computed separately at each sensor on the scalp. Averages are computed from the moment of the potential talker switch to the end of the trial. The bottom bar graphs show the average normalized alpha change over the posterior left and posterior right channels computed from the scalp distributions. For switching trials, parietal alpha lateralization is weak after the talker change.

by its location, alpha lateralization appears only transiently at the start of a trial if competing streams have distinct nonspatial features (*Bonacci et al., 2019*). Thus, spatial auditory attention utilizes parietal cortex and produces lateralized alpha; however, nonspatial auditory attention does not and produces no signature of attentional focus in parietal cortex.

## Talker discontinuity disrupts auditory spatial attention

Numerous behavioral studies demonstrate that maintaining attention on an ongoing auditory stream is supported by continuity of features like pitch, location, voice, and timbre (*Best et al., 2008*; *Darwin and Carlyon, 1995*). Feature continuity influences performance automatically: even when listeners know they should ignore some feature, such as talker identity, and attend to a different feature, such as location, discontinuities in the task-irrelevant feature disrupt attention (*Best et al., 2008*; *Maddox and Shinn-Cunningham, 2012*; *Bressler et al., 2014*). Indeed, effects of talker continuity on speech perception in quiet have been ascribed to 'talker normalization,' but may actually

be due to disruptions of attention triggered by a talker change (*Lim et al., 2019*; *Choi and Perrachione, 2019*).

Talker discontinuity during an auditory spatial attention task not only interferes with recall of the target stream, it disrupts parietal alpha lateralization (*Mehraei et al., 2018*) (see *Figure 1B*). In this recent study, two competing speech streams were presented. Each syllable was presented from either left or right and was spoken by either a male or a female. Subjects were cued at the beginning of a trial to focus on and report back the content of either the left or the right syllables. The cue conveyed only to which location the subject should attend. In *continuous* trials, the talker from each direction was fixed throughout the trial, while in *switching* trials, the talkers switched locations after the first two target syllables (see top of *Figure 1B*). Because the *continuous* and *switching* trials were randomly interweaved within experimental runs, subjects had to rely on top-down spatial attention to perform the task even in the *continuous* trials. In *switching* trials when listeners were instructed to listen to syllables from one direction (ignoring any talker change), errors were elevated for the syllable right after the talker switch —and subsequent alpha lateralization was disrupted (see bottom of *Figure 1B*). These results suggest that talker continuity overrides top-down, volitional control of spatial attention; when the talker from the attended location jumps to the opposite side of the listener, the talker discontinuity interrupts spatial attention and parietal alpha lateralization.

## Brain stimulation studies suggest a causal link between parietal alpha and visual spatial attention

Despite the strong association between alpha lateralization and spatial visual and auditory attention, these results do not prove that parietal alpha 'steers' attention. In humans, brain stimulation methods such as transcranial magnetic stimulation (TMS) and transcranial electrical stimulation (for reviews, see *Bestmann et al., 2015*; *Dayan et al., 2013*; *Herrmann et al., 2016*; *Parkin et al., 2015*) provide a means to directly test whether particular neural regions are causally involved in particular behaviors.

TMS inactivation of parietal cortex in one hemisphere causes spatially specific visual processing changes, enhancing spatial attention directed ipsilaterally but degrading it contralaterally (*Hilgetag et al., 2001*). Unilateral TMS inactivation of the frontal eye field (FEF, part of the fronto-parietal visuo-spatial attention network) reduces alpha coupling between prefrontal and parietal cortices; moreover, parietal alpha lateralization is disrupted and this disruption predicts increases in reaction times during a visual working memory task (*Sauseng et al., 2011*). Unilateral 20 Hz rTMS (which disrupts alpha oscillations) of either FEF or parietal cortex has similar effects (*Capotosto et al., 2009*). These stimulation studies confirm that the fronto-parietal network is involved in controlling spatial attention, yet still beg the question: are alpha oscillations causally responsible for suppressing contralateral information, or are they an epiphenomenon?

Some studies have demonstrated effects of alpha-rate stimulation of parietal cortex on visual perception using repetitive TMS (rTMS; *Klimesch et al., 2003*) or transcranial alternating current stimulation (tACS; *Helfrich et al., 2014*); however, only a handful have addressed whether such stimulation affects spatial processing. Alpha-rate rTMS of parietal cortex enhances performance for ipsilateral targets and degrades performance for contralateral targets for visual spatial attention (*Romei et al., 2010*) and working memory (*Sauseng et al., 2009*) tasks, while stimulation at non-alpha frequencies has no effect. Yet, tACS results are equivocal. Two studies failed to find frequency- or retinotopically specific effects of parietal tACS stimulation on visual tasks (*Brignani et al., 2013*; *Veniero et al., 2017*), while a high-density tACS (HD-tACS) found that parietal alpha stimulation affects volitional control of visual spatial attention, improving performance for targets ipsilateral to the stimulation (*Schuhmann, 2019*).

A few studies have shown that stimulation of auditory cortex can influence auditory task performance (*Riecke et al., 2018*; *Riecke et al., 2015*; *Zoefel et al., 2018*; *Neuling et al., 2012*), including spatially specific effects on auditory selective attention (*Wöstmann et al., 2018*; *Hanenberg et al., 2019*). However, we are unaware of any prior studies exploring whether parietal alpha stimulation influences auditory spatial attention.

## Rationale of the current study

Compared to traditional tACS (conducted through sponge pads), HD-tACS, which uses an electrode ring configuration (e.g., an anode flanked by multiple cathodes), creates a more focused electrical current sink (*Kuo et al., 2013*; *Reinhart and Nguyen, 2019*). This allows for more precise anatomical targeting. Combined with improved computational models of predicted current flow in the brain (*Datta et al., 2009*; *Edwards et al., 2013*), HD-tACS yields more precise brain stimulation than traditional approaches. We therefore used HD-tACS to achieve focused, alpha-frequency stimulation of parietal cortex.

Our goal was to show that parietal alpha causally affects performance on a spatial auditory task in a hemisphere-specific manner. Listeners focused attention on a stream of syllables while ignoring a similar, competing stream. Our experimental design included two main conditions for which we had a priori hypotheses about the direction of an effect, which are described below. We also included multiple levels of controls designed to test the specificity of stimulation effects on performance, where we did not expect to see effects of stimulation. Specifically, we expected alpha stimulation to only impact conditions where listeners volitionally focused and could maintain spatial attention (and alpha lateralization).

Listeners performed the same basic task of focusing on a stream of spoken syllables, but focused either on location (spatial attention, where parietal alpha HD-tACS should modulate performance) or talker gender (nonspatial attention, where parietal stimulation should have no impact; see *Figure 2A*). On half of the trials the talker from a particular direction remained fixed (*continuous* trials) and in the other half, the talker alternated from syllable to syllable (*switching* trials; see *Figure 2B*). As discussed above, a sudden spatial shift of talkers interferes with top-down spatial attention and disrupts alpha lateralization (*Mehraei et al., 2018*) (*Figure 1B*). Therefore, we expected alpha stimulation to have no effect on *switching* trials, as talker discontinuities should exogenously disrupt spatial attention and parietal alpha. We hypothesized that on continuous trials listeners would normally be able to focus spatial attention through alpha lateralization, so alpha stimulation would influence performance.

Subjects were instructed and explicitly aware that in both the spatial and talker attention blocks, each trial was equally (and unpredictably) likely to be a switching trial or a continuous trial. They had no way of predicting which trials would be switching and which continuous; therefore, there was no way for them to have adopted different listening strategies for the continuous and the switching trials within a block. Furthermore, they were aware (and explicitly instructed) that attending to voice was not a reliable strategy in the attend-location block.

We stimulated parietal cortex unilaterally, testing for spatial specificity of the stimulation. We targeted right intraparietal sulcus (rIPS) based on previous findings suggesting that right parietal cortex contains the only representation of left exocentric space, whereas right space is represented strongly in left IPS but also weakly in rIPS (*Farah et al., 1989*; *Thiebaut de Schotten et al., 2011*; *Shulman et al., 2010*). During spatial attention, we expected rIPS alpha stimulation to suppress the representation of left exocentric space, impairing performance for leftward targets (see *Figure 3*, bottom left panel). We had a secondary hypothesis that performance for rightward targets might either be unchanged (as rIPS should already have strong alpha due to top-down spatial attention) or enhanced (if stimulation led to even better suppression of the leftward distractor; see *Figure 3*, bottom right panel).

Each subject performed two full sessions on separate days (order counter-balanced across subjects). The Sham session applied transient currents to convince subjects that they were being stimulated, while true HD-tACS was applied in the Stimulation session (see *Figure 2C*). The Sham sessions thus provided a direct within-in subject control for the Stimulation sessions. Similarly, each session began with a block of no-stimulation baseline trials, then presented a block of trials with either HD-tACS or sham stimulation, and finally finished with a block of trials with no stimulation (*Figure 2C*). We expected effects of HD-tACS stimulation to appear during the middle, 'stimulation' block, but to dissipate quickly, with performance in the no-stimulation block returning to baseline.

Finally, to investigate frequency specificity, we conducted two experiments differing only in the frequency of HD-tACS stimulation during the Stimulation session. In the main experiment, we used alpha (10 Hz) stimulation. We then performed a final control experiment that used theta (6 Hz) stimulation. We chose theta as a control frequency because, like alpha, theta is an intrinsic oscillation

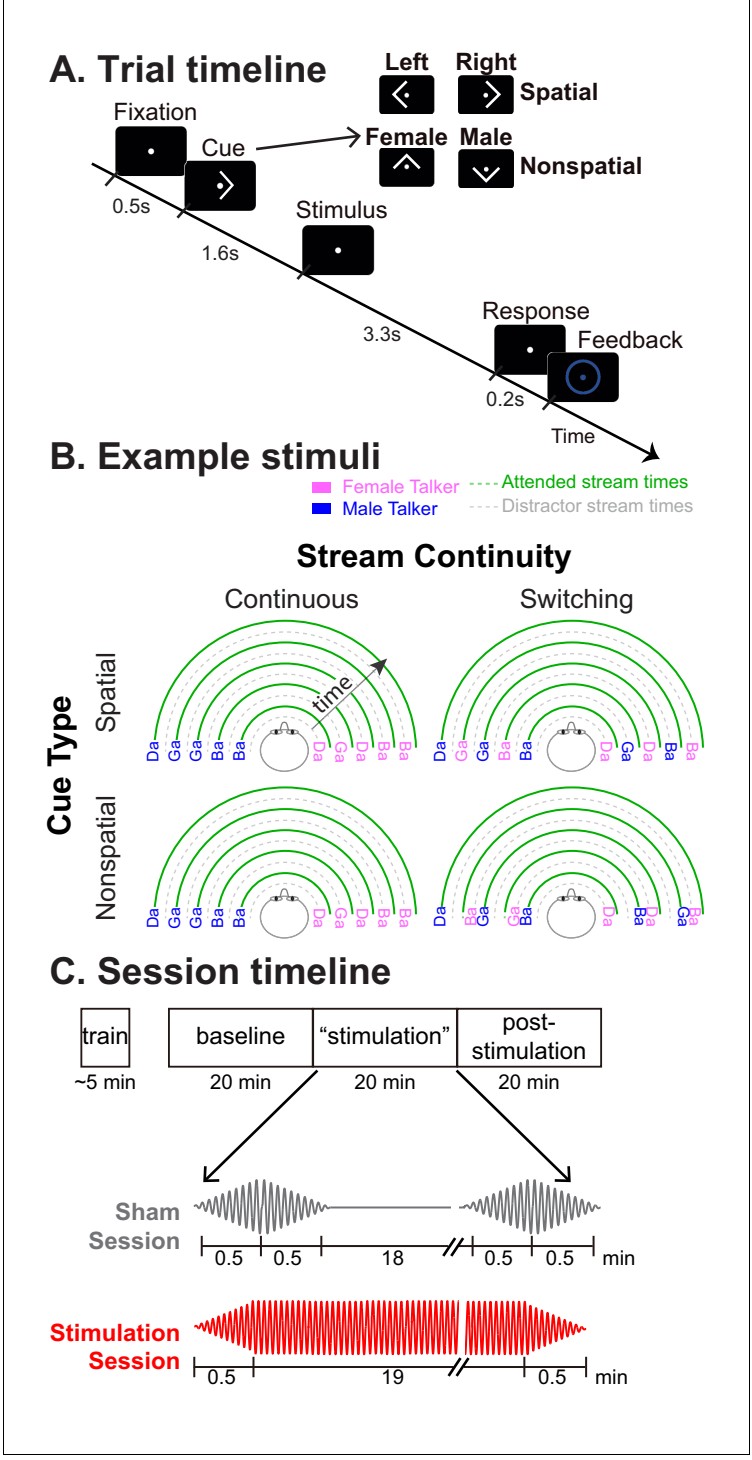

**Figure 2.** Task paradigm. (**A**) Trial timeline. Subjects were instructed to foveate on a fixation dot appearing at the start of the trial. A visual cue then appeared, instructing them how to focus attention in the upcoming trial: left or right (a spatial trial) or male or female talker (a nonspatial trial). The target and distractor streams then began to play. Subjects were instructed to count the number of /ga/ syllables in the 5-syllable target stream. After the auditory stimulus, they were asked to report this count as quickly as possible. Following their response, a circle appeared around the fixation dot indicating whether they were correct or incorrect. (**B**) Diagrams of example stimuli for the different trial types. The first syllable always was a distractor. Syllables were temporally interdigitated, alternating between distractor and target, in a temporally regular pattern. In continuous trials, the talker from each direction was fixed throughout the trial. In switching trials, the talker from one direction

*Figure 2 continued on next page*

*Figure 2 continued*
alternated from syllable to syllable. (**C**) Session timeline. Following a brief training session, subjects performed three 20-min-long blocks of trials: pre-stimulation baseline, 'stimulation,' and post-stimulation. In Sham sessions, HD-tACS was ramped on and off at the start and end of the 'stimulation' block to blind subjects as to the condition. In Stimulation sessions, HD-tACS ramped up at the start and down at the end of the 'stimulation' block.

occurring in parietal cortex (*Jensen and Tesche, 2002*; *Raghavachari et al., 2001*), but one that is not linked to spatial attentional control. Theta also closely neighbors the alpha band, making it a stringent control. We hypothesized that alpha, but not theta, stimulation would affect spatial attention performance.

Our primary interest was to explore how HD-tACS stimulation influenced spatial attention. Amongst all of the conditions tested, there were two specific cases in the main experiment where we predicted a difference in performance between the Sham and tACS sessions, each of which we expected to have a particular direction; the other conditions were included solely as controls. Specifically, during stimulation when listeners were using spatial attention for continuous stimuli, we expected performance to be 1) worse in tACS than Sham sessions for leftward attention, but 2) either better (or unchanged) in tACS than Sham sessions for rightward attention. We therefore

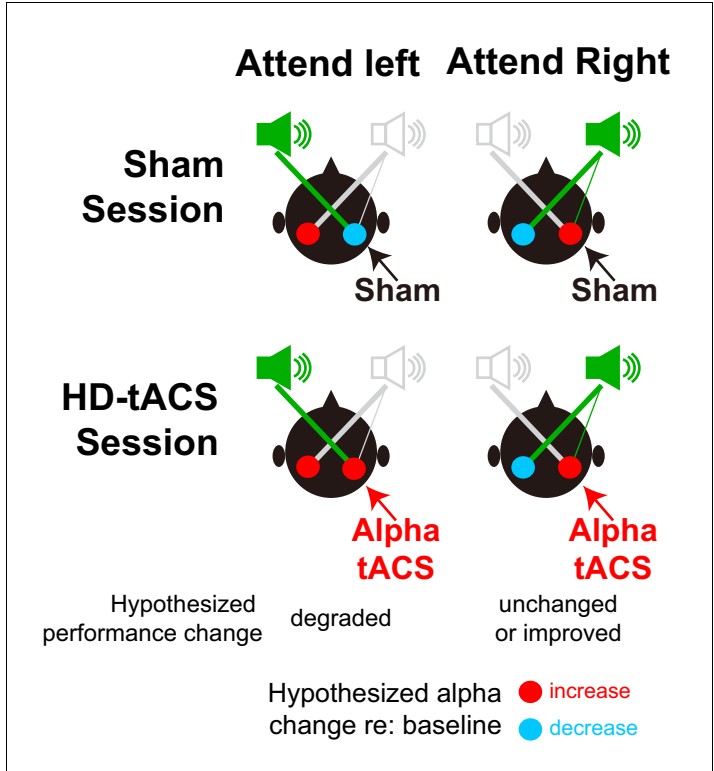

**Figure 3.** Hypothesized alpha power in left and right parietal hemispheres, relative to baseline, during spatial attention trials. The top row shows the expected patterns in Sham sessions, while the bottom row shows the hypothesized effects of HD-tACS alpha stimulation of rIPS. Information is asymmetrical represented; information from the left is represented only in the right (contralateral) parietal hemisphere, while information from the right is represented most strongly in the left (contralateral) hemisphere, but also weakly in the right hemisphere. In the absence of any stimulation (top row), top-down attention should lead to a decrease in alpha power in the hemisphere contralateral to the direction of attention (allowing the dominant representation of the attended location to be processed) and an increase in the ipsilateral hemisphere (suppressing the dominant representation of the ignored location). We hypothesized that applying alpha HD-tACS to rIPS should suppress the representation of leftward space, interfering with processing of left targets (bottom left). However, alpha stimulation of rIPS should either have little effect, or perhaps enhance processing of rightward targets, as rightward top-down attention already produces strong alpha in rIPS (bottom right).

planned to conduct two signed planned comparisons, for these two conditions, a priori. We did not expect any effects of theta stimulation in the control experiment. Incidentally, we expected to replicate previous results showing that talker switching interferes with spatial attention, a question we addressed by comparing performance on continuous and switching trials for the initial block of trials across all sessions, before stimulation.

## Results

### Results confirm that talker switches exogenously interfere with spatial attention

Based on previous results (*Mehraei et al., 2018*), we expected performance to be worse in *switching* than *continuous* trials, especially during spatial attention. Results confirmed this (see *Figure 4*). We averaged performance for the baseline blocks of both the Sham and Stimulation sessions, since these blocks were identical, occurring prior to any stimulation. We expected performance for these baseline blocks to be similar in the main experiment and the control experiment, since the trials in these blocks were identical across the experiments (though the subjects differed). As seen in *Figure 4*, performance was worse in *switching* trials than in *continuous* trials, especially during spatial attention, in both experiments.

Results from the main experiment are shown in *Figure 4A*. When listeners attended to the left, performance changed in the expected direction, dropping from 87.40% correct in *continuous* trials to 65.29% in *switching* trials ($Z_{(19)} = 3.91$, $P_{adj} < 0.001$, Wilcoxon signed rank test). When listeners attended to the right, performance also dropped, from 88.27% for *continuous* to 64.81% for *switching* trials ($Z_{(19)} = 3.90$, $P_{adj} < 0.001$). When listeners directed attention to a specific talker, the average accuracy also dropped slightly for *switching* compared to *continuous* trials for both attend-female (79.62% to 79.13%) and attend-male (85.48% to 81.44%) trials. For attend-female trials, this change was not significant ($Z_{(19)} = 0.20$, $P_{adj} > 0.99$); however, it did reach significance for attend-male trials ($Z_{(19)} = 2.36$, $P_{adj} = 0.036$).

Results from the control experiment, shown in *Figure 4B*, were similar. When listeners attended to the left, performance dropped as expected, from 79.81% correct in *continuous* trials to 56.84% correct in the *switching* trials ($Z_{(17)} = 3.66$, $P_{adj} < 0.001$, Wilcoxon signed rank test corrected for multiple comparisons). When listeners attended to the right, performance dropped from 80.77% to 57.26% ($Z_{(17)} = 3.71$, $P_{adj} < 0.001$). When listeners directed attention to a specific talker, average

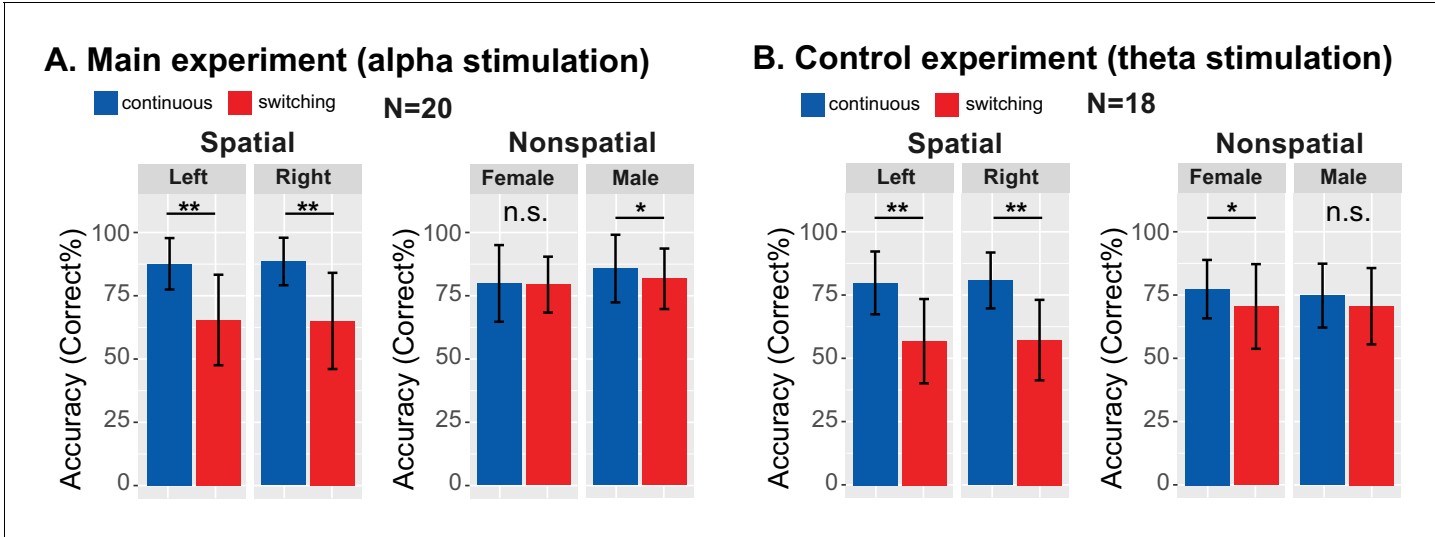

**Figure 4.** Behavioral performance averaged across baseline blocks in which there was no stimulation. Talker switching significantly disrupts spatial attention, but not nonspatial attention. (A) Results from the main experiment. The left plot shows results for spatial attention, comparing continuous and switching trials; the right plot shows results for nonspatial attention. Error bars represent the across-subject standard error of the means. Double asterisks indicate statistical differences with p<0.001. (B) Results from the control experiment, laid out as in (A).

accuracy was lower for *switching* than *continuous* trials for both attend-female (77.35% to 70.51%) and attend-male (74.79% to 70.62%) trials. This change reached statistical significance for the attend-female trials ($Z_{(17)}$ = 2.76, $P_{adj}$ = 0.011), but not the attend-male trials ($Z_{(17)}$ = 1.86, $P_{adj}$ = 0.13).

It is worth noting that our talkers switched after each syllable (see *Figure 2B*), which should be more disruptive than a single switch (as in *Mehraei et al., 2018*, which inspired this manipulation). Consistent with this, we found a larger drop in performance from *continuous* to *switching* trials than in *Mehraei et al. (2018)*.

## Baseline performance is similar in sham and stimulation sessions

Each subject in both experiments performed both a Sham and an HD-tACS Stimulation session. Session order was counter-balanced across subjects, who were blinded to this aspect of the experimental design. In both Sham and Stimulation sessions, the first trial block was a no-stimulation, baseline block.

We first confirmed that there was no significant difference in baseline performance between Sham and Stimulation sessions in either the main experiment ($Z_{(19)}$ = 0.068, p=0.95, Wilcoxon rank test) or the control experiment ($Z_{(17)}$ = 0.46, p=0.65, Wilcoxon rank test). Thus, counter-balancing the session order canceled out any systematic effects of testing order.

To correct for changes in individual performance between sessions, we referenced performance to that in the baseline block in each session. To test whether our results could be confounded by performance fluctuations in this baseline level between test days, we performed test-retest reliability analyses, comparing baseline results across different testing days (*Reinhart, 2017*). We found that individual subjects' performance across sessions was significantly correlated in both the main experiment (Spearman's Rhos = 0.48; p=0.032) and the control experiment (Spearman's Rhos = 0.52; p=0.028), indicating the stability of individual differences across testing days.

## Alpha HD-tACS of rIPS disrupts auditory spatial attention for leftward targets

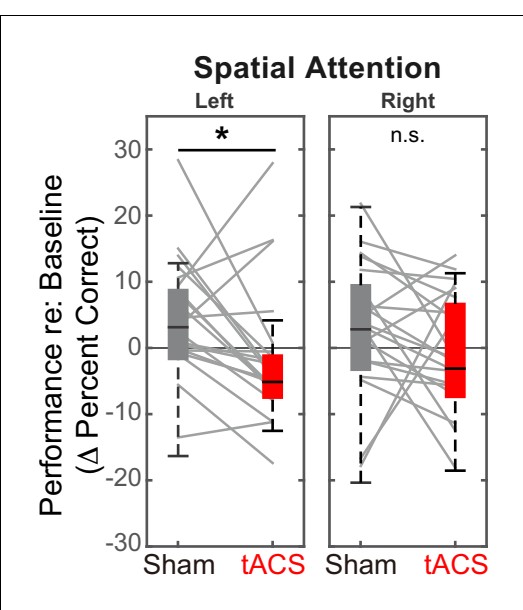

**Figure 5.** Baseline-corrected performance for spatial attention trials in the 'stimulation' blocks, comparing Sham and Stimulation sessions. Parietal HD-tACS stimulation shows spatially specific and frequency-specific effects. Compared to Sham stimulation, rIPS alpha HD-tACS stimulation significantly impaired performance on attend-left trials, but had no effect on attend-right trials.

We conducted within-subject comparisons of baseline-corrected performance in Sham and Stimulation sessions for trials where we expected an effect. We analyzed only *spatial* trials, since parietal processing is not involved during non-spatial attention, and only *continuous* trials, where there was no exogenous disruption of spatial attention. Finally, we expected any effects to be present during, but not after stimulation. Thus, a priori, we considered only two of the many conditions tested; the control trials from the main experiment and all the results from the control experiment are shown in the next section.

We predicted stimulation to decrease performance for leftward spatial attention compared to the Sham session, but either to increase or to have no effect for rightward spatial attention. Results confirmed these expectations (see *Figure 5A*). Specifically, for rIPS alpha stimulation, performance in the 'stimulation' block was significantly worse in the Stimulation session than in the Sham session for left attention ($Z_{(19)}$ = 2.10, $P_{adj}$ = 0.036, Wilcoxon signed rank test, corrected for multiple comparisons). The effect size of stimulation on leftward attention was 0.33, computed using the *z* value obtained from

the Wilcoxon test with the formula: $d = z/\sqrt{N}$ (*Pallant, 2013*). There was no significant increase in performance from Sham to Stimulation sessions for right attention ($Z_{(19)}$ = 1.27, $P_{adj}$ > 0.99; see *Figure 5A*).

Post hoc, we explored the dynamics of the effect of alpha stimulation on spatial attention to leftward targets. For both the Sham and the Stimulation sessions, we subdivided both the stimulation block and the post-stimulation block into four sub-sessions (each comprising six spatial, attend-left trials) and computed the baseline-corrected performance for each (*Figure 6*).

We observed a consistent, sustained effect of stimulation: baseline-corrected performance was lower in the Stimulation session compared to the Sham session for each of the sub-blocks in the 'stimulation' block (left side of *Figure 6*). This difference disappeared by the first post-stimulation sub-block, immediately after HD-tACS stopped; baseline-corrected performance was indistinguishable for the Sham and Stimulation sessions for the final four sub-blocks (right side of *Figure 6*).

## Theta, stimulation does not affect performance during spatial attention

*Figure 5A* demonstrates that there is a spatially specific effect of HD-tACS stimulation. We included a number of other control conditions where we expected no effects of stimulation (see *Figure 7*).

Theta stimulation was not expected to alter parietal processing for any trials (all panels in *Figure 7B*). Because parietal cortex should not be strongly engaged during nonspatial attention, we expected no stimulation effects in any of the nonspatial attention trials (right half of *Figure 7A and B*). We expected any effects of stimulation in spatial attention trials to dissipate rapidly, with no residual effect in the post-stimulation block (bottom rows in *Figure 7A and B*). Because talker switches exogenously disrupt spatial attention (and likely parietal alpha), we expected no influence of stimulation during *switching* trials, even in spatial-attention trials (third and fourth panels of the top row in *Figure 7A and B*). These expectations were all borne out by our results.

Of the 32 distinct trial types, Sham vs. Stimulation sessions differed significantly only in one, in the expected direction: during alpha stimulation of rIPS, when listeners directed spatial attention to

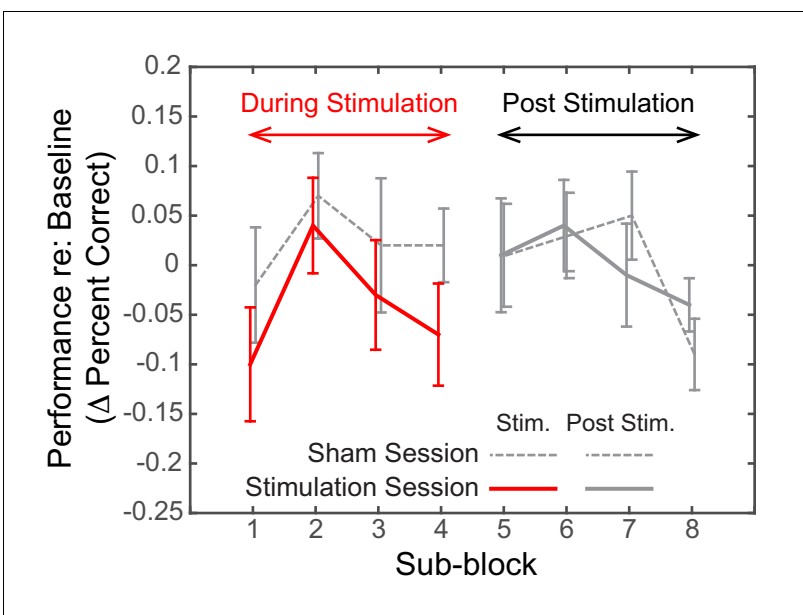

**Figure 6.** Dynamics of the effect of rIPS alpha HD-tACS stimulation on spatial, attend-left trials, comparing Sham and Stimulation sessions. Performance is consistently poorer for alpha stimulation than for sham stimulation throughout the course of stimulation; however, performance in the sessions is indistinguishable once stimulation ends. Trials within both the 'stimulation' block and post-stimulation block were divided into four sub-blocks each. Baseline-corrected performance and the standard error of the mean across subjects are shown for the Sham and the Stimulation sessions (dashed and solid lines, respectively). Data from the 'true' stimulation sub-blocks are shown in red (the first four sub-blocks of the Stimulation session); no-stimulation sub-blocks are shown in gray (all sub-blocks of the Sham session, as well as the final four sub-blocks of the Stimulation session).

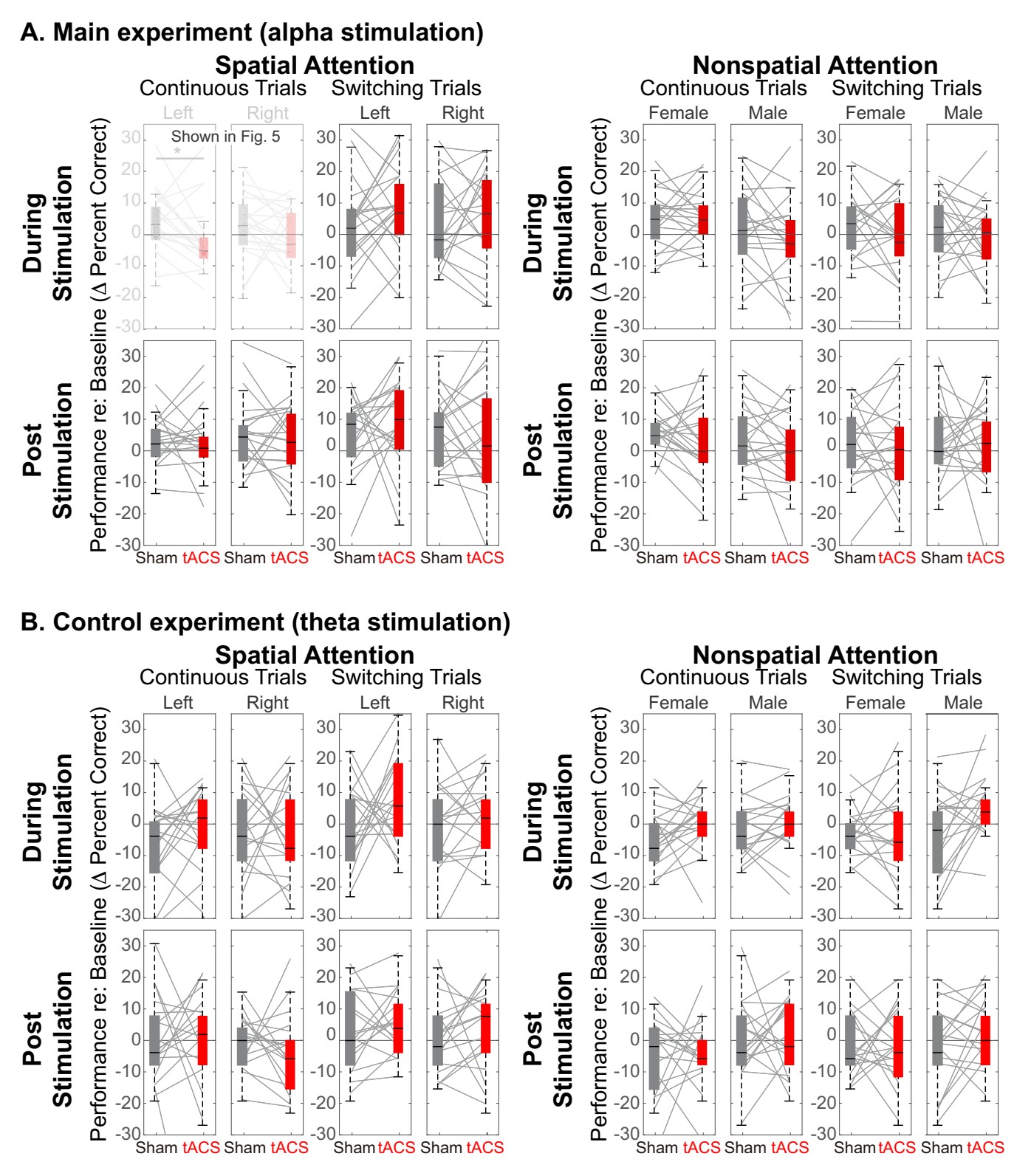

**Figure 7.** Comparison of Sham and Stimulation session results. There is no significant effect of stimulation in any of the control conditions. (Note that the top left panels in (A) are repeated from *Figure 5*). (A) Results from the main experiment, using alpha HD-tACS. Results for spatial attention are shown on the left, and for nonspatial attention shown on the right. For each form of attention, results are broken down into continuous and switching trials (two leftmost and two rightmost panels, respectively). For spatial attention, results are shown for both attend-left and attend-right trials; for

*Figure 7 continued on next page*

*Figure 7 continued*

nonspatial attention, results are shown for attend-female and attend-male trials. Finally, results from trials during the 'stimulation' block are shown in the top row and from the subsequent post-stimulation block in the bottom row. (B) Results from the control experiment, using theta stimulation, laid out as in (A).

a continuous-talker target that was on the left (top leftmost panel in *Figure 7A*, repeated from *Figure 5A*). *Table 1* shows the results of statistical tests directly comparing Sham and Stimulation baseline-corrected results for all trial types in the main experiment (Wilcox rank tests; note that for control conditions, there was no correction for multiple comparisons, which is conservative given that we expected null results). In addition, there was no effect of theta stimulation for any of the 16 trial types in the control experiment [(spatial and nonspatial attention) x (during stimulation and post stimulation) x (continuous trials and switching trials) x (left/female and right/male)]: for all of these, $Z_{(17)} < 1.67$ (p>0.095).

We did one final post hoc exploration of the data from the main experiment, examining performance for the *continuous* nonspatial trials, which used stimuli that were physically identical to those where we saw stimulation effects (*continuous* trials during spatial attention). We reanalyzed performance for the *continuous* nonspatial trials based on the direction of the target (a feature that listeners were presumably ignoring), collapsing across the gender of the target talker (the feature that listeners were presumably attending). We then examined performance for leftward and rightward targets. If parietal stimulation disrupted attention for leftward, continuous targets when attention was directed to talker gender, it would suggest that spatial parietal processing contributed to performance even during nonspatial attention trials. Comparing baseline-corrected performance for Sham vs. Stimulation sessions during nonspatial attention, we observed no effect of stimulation for targets from either the left ($Z_{(19)} = 0.23$, p=0.82) or the right ($Z_{(19)} = 1.30$, p=0.19). These results further support the view that alpha stimulation of parietal cortex only influences top-down spatial processing.

## Discussion

### Summary of results

Using HD-tACS to stimulate rIPS, we here show a causal relationship between parietal alpha power and spatial auditory attention. We included numerous controls to establish the specificity of the influence of parietal alpha. We used a within-subject design that included counter-balanced Sham and Stimulation sessions, and that employed both baseline and post-stimulation control blocks within each session to control for performance fluctuations and to validate the internal consistency of the behavioral metrics. We found that right HD-tACS parietal stimulation disrupted selective attention directed to the left. We found no effect on performance in the myriad control conditions (i.e., when

**Table 1.** Statistical tests comparing baseline-corrected performance in Sham and Stimulation sessions for the various types of control trials in the main experiment, laid out as in *Figure 7A*.

Each cell represents results of a Wilcoxon rank test with 19 degrees of freedom. Note that the primary comparisons of interest (left vs. right spatial attention for continuous stimuli during rIPS alpha stimulation; results shown in light gray) were already discussed in Section 2.3. For these conditions, results are corrected for multiple comparisons. Uncorrected statistics are reported for the remaining control conditions.

**Main experiment, using alpha stimulation**

| | Spatial attention | | | | Nonspatial attention | | | |
| | Continuous | | Switching | | Continuous | | Switching | |
| | Left | Right | Left | Right | Female | Male | Female | Male |
|---|---|---|---|---|---|---|---|---|
| During Stimulation | Z = 2.10 $P_{adj}$ = 0.036 | Z = 1.27 $P_{adj}$ = 1 | Z = 1.21 p=0.23 | Z = 0.58 p=0.56 | Z = 0.11 p=0.91 | Z = 1.08 p=0.28 | Z = 0.69 p=0.49 | Z = 0.71 p=0.48 |
| Post Stimulation | Z = 0.87 p=0.39 | Z = 0.23 p=0.82 | Z = 1.03 p=0.30 | Z = 0.66 p=0.51 | Z = 1.02 p=0.31 | Z = 0.98 p=0.33 | Z = 0.33 p=0.74 | Z = 0.10 p=0.92 |

listeners focused top-down attention based on talker identity, when the talker in the attended direction switched abruptly, or when theta stimulation was employed).

In our study, alpha HD-tACS stimulation produced an effect size of 0.33 for leftward spatial attention. A meta-analysis study (*Schutter and Wischnewski, 2016*) shows that this effect size is comparable to those in past published reports of tACS effects on cognitive function in healthy adults ($\bar{E}$=0.36, 95% *CI* = 0.27–0.46) as well as tDCS studies (e.g., $\bar{E}$ = 0.23, 95% *CI* = 0.09–0.36) (*Brunoni and Vanderhasselt, 2014*; *Hill et al., 2016*; *Klaus and Schutter, 2018*). Thus, we not only established a very specific role of parietal alpha in auditory selective attention, the effect is as robust as other reported effects of neural stimulation.

## Spatial and frequency specificity show that parietal alpha causally influences auditory spatial attention

Our main experiment used HD-tACS in the alpha band (10 Hz), while in our control experiment, stimulation was at a closely neighboring theta frequency (6 Hz). We found no evidence that theta stimulation altered performance. This frequency specificity of HD-tACS parietal stimulation implicates alpha in control of auditory spatial attention. As expected, we found that stimulation interfered with spatial attention to leftward sources, but had no significant effect for rightward sources (see *Figure 3*).

We observed no improvement for rightward spatial attention, even though we thought that right parietal stimulation might enhance suppression of an interfering leftward stream. This lack of an effect could simply be due to limited statistical power. However, it may also reflect other factors. First, previous visual attention studies show that alpha-tACS increases endogenous alpha oscillations only when alpha power is low (*Alagapan et al., 2016*; *Neuling et al., 2013*). Top-down attention to the right should naturally produce strong rIPS alpha, limiting the influence of additional alpha power (see *Figure 3*). In contrast, intrinsic rIPS alpha power should be low during leftward attention, allowing stimulation to have an observable effect. Second, parietal cortex is asymmetrical; rIPS predominantly represents left space, but also has a weak representation of right space. When listeners attended to the right, an injection of alpha energy to rIPS may have enhanced suppression of the dominant leftward distractor, but it may also have spread to suppress the weak representation of the rightward target, leading to little net change. Finally, as discussed further below, our stimulation may not have been precise enough to optimize its effectiveness. Regardless, the spatial specificity of the effect of rIPS stimulation is consistent with the hypothesis that parietal alpha causally suppresses the representation of contralateral space, steering spatial attention.

## Task-specificity and stimulus-specificity show that parietal alpha stimulation influences volitional spatial attention, but not exogenous attention

Consistent with our hypotheses, we found no effect of parietal stimulation when listeners directed attention based on talker gender. To further check that rIPS stimulation did not impact nonspatial attention, we analyzed nonspatial attention performance for *continuous* targets that happened to be from the left – physically identical to the stimuli for which rIPS HD-tACS alpha stimulation impaired spatial attention performance. As expected, HD-tACS stimulation had no significant impact on nonspatial attention.

In *switching* stimuli, the irrelevant target feature alternated from syllable to syllable. Based on previous results (*Mehraei et al., 2018*), we postulated that talker switches would cause exogenous, involuntary disruptions of spatial attention and of parietal alpha lateralization. We further expected these exogenous interruptions to override top-down, parietal influences on spatial focus, rendering parietal stimulation irrelevant. Behaviorally, we verified that *switching* stimuli impaired spatial attention performance in the absence of stimulation. We also found, as expected, no evidence that alpha stimulation affects perception when parietal alpha lateralization is already disrupted by talker discontinuities.

Thus, our results implicate parietal alpha in volitional steering of auditory spatial attention.

## Our results confirm and clarify past results from visual attention studies

As noted in the Introduction, past studies exploring how parietal alpha stimulation influences performance have produced apparently conflicting results. Alpha rTMS of parietal cortex increases performance for ipsilateral targets and decreases performance for contralateral targets, both in visual attention and visual working memory tasks, while no effects are found for other stimulation rates (*Romei et al., 2010*; *Sauseng et al., 2009*). However, past results from the handful of studies that used tACS to stimulate parietal cortex during visual spatial attention are less conclusive.

One study directly compared unilateral sham, 6 Hz, 10 Hz, and 25 Hz tACS stimulation of parietal cortex (*Brignani et al., 2013*), but found no retinotopic specificity of stimulation and only weak frequency specificity during a visual detection task. Another tACS study found that right parietal stimulation shifted the perception of the midpoint of a line segment while sham stimulation did not; however, a follow up experiment in the same study failed to replicate the initial finding, with no significant effect of sham or alpha stimulation (*Veniero et al., 2017*). While these two studies seem to suggest that lateralized parietal alpha may not causally steer visual spatial attention, a more recent study offers a more nuanced explanation. Schumann and colleagues (*Schuhmann, 2019*) compared the effects of HD-tACS alpha stimulation of left parietal cortex for three visual tasks: a *detection* task, an *exogenous* spatial attention task, and an *endogenous* spatial attention task. Stimulation had no effect on *detection* of a faint visual grating. In the spatial attention tasks, observers had to not just detect, but also report the orientation of the grating. In the *exogenous attention* task, four dots appeared around one of the potential target positions and the target either appeared in that position (congruent; 50% of the trials) or in the opposite hemifield (incongruent; 50% of the trials). Observers were better in congruent than incongruent trials—but, critically, parietal stimulation had no impact on performance. Finally, in the *endogenous attention* task, a visual cue correctly indicated the location of a subsequent target on 80% of the trials, providing a top-down cue for spatial attention. In this case, and only this case, parietal alpha stimulation caused a spatially specific effect, decreasing reaction times for ipsilateral targets.

These findings highlight the importance of carefully considering task demands when interpreting results of parietal stimulation studies. While parietal alpha modulates volitional control of spatial attention tasks, it does not robustly influence *exogenous* attention. The sudden appearance of a new stimulus, even one near threshold, may draw exogenous attention (*Desimone and Duncan, 1995*), which may override any effects of parietal processing and render alpha parietal stimulation impotent (*Brignani et al., 2013*).

In addition, whereas Schumann and colleagues used HD-tACS, the studies that failed to see consistent, spatially specific effects of parietal alpha stimulation used traditional tACS. Traditional tACS is usually delivered with large pads ($5 \times 5$ cm$^2$) and stimulates a broad area between the stimulation electrodes (*Datta et al., 2009*). The resulting spread of electric current is greater, and could even spread to both hemispheres, confounding stimulation effects.

## Our study differs from past brain stimulation studies in audition

As noted in the Introduction, a few studies stimulated auditory cortex and demonstrated behavioral effects (*Riecke et al., 2018*; *Riecke et al., 2015*; *Zoefel et al., 2018*). However, we know of no other studies that show a causal influence of parietal alpha oscillations on auditory spatial attention.

The most closely related study used traditional tACS to target a large region of left hemisphere that included portions of temporal and inferior parietal cortices (*Riecke et al., 2015*). This study shows a double dissociation between stimulation at alpha vs. gamma frequencies; specifically, alpha stimulation degrades attention to contralateral stimuli, while gamma stimulation improves contralateral attention.

In contrast to their study, our HD-tACS stimulation targeted intraparietal sulcus and produces essentially no current in auditory sensory regions (see *Figure 8C*). The two studies are consistent in showing that alpha stimulation impairs attention to contralateral auditory space. However, ours demonstrates that alpha in IPS, which is a part of the well-studied visuo-spatial attention network, plays a causal role in spatial auditory attention, whereas the effects reported by Wöstmann and colleagues could be due to stimulation of auditory sensory regions. By including a nonspatial attention task as a control, our study further shows that the influence of IPS alpha depends specifically on top-down engagement of the visuo-spatial attention network; there is no influence of IPS alpha stimulation

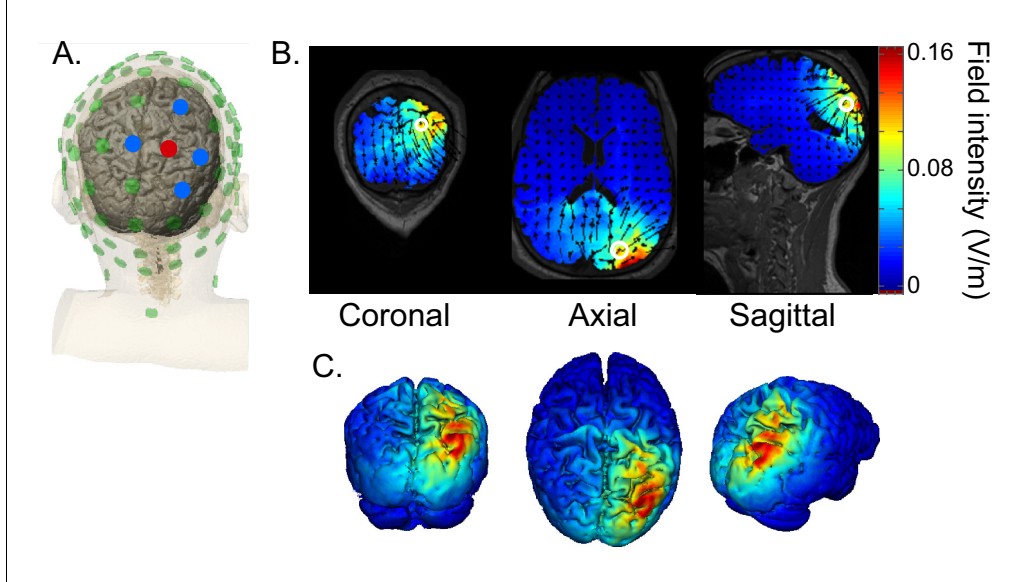

**Figure 8.** HD-tACS model, targeting rIPS. (**A**) Diagram of the electrode montage. Five parietal channels (CP2, P2, P4, Pz, POz) were selected from a standard 64-channel EEG cap. The blue and red dots, indicating opposite polarities, show the chosen stimulation polarity distribution. This distribution produces a current sink targeting right intraparietal sulcus (IPS). (**B**) The current flow model based on the selected montage, showing 2D coronal, axial and sagittal views of the stimulation reaching each position in the brain. The open white circle indicates the targeted Montreal Neurological Institute coordinate (24, -70, 41), based on a previous fMRI study that mapped the location of human IPS (*Swisher et al., 2007*). (**C**) Three views of the 3D model simulation.

when listeners attend to the same auditory source within the same physical sound mixture using non-spatial auditory features.

We believe that auditory spatial attention recruits the same fronto-parietal network involved in visual spatial attention (*Michalka et al., 2016*; see the discussion in *Bonacci et al., 2019*). We therefore expect manipulation of parietal alpha to lead to similar effects for auditory spatial attention and visual spatial attention. Consistent with *Schuhmann (2019)*, we used alpha HD-tACS to focally stimulate parietal cortex. As in *Schuhmann (2019)*, we found that manipulation of parietal alpha affects control of *endogenous* spatial attention (processing of *continuous* contralateral targets during a spatial attention task)—but does not influence performance dominated by *exogenous* attention effects (processing of *switching* targets, where *endogenous* disruptions limit performance).

## Caveats and future work

We know of no other studies that show a causal influence of parietal alpha oscillations on auditory spatial attention. Additional studies are needed to replicate and confirm our results, and to further delineate the conditions under which parietal alpha influences auditory perception.

We used HD-tACS to achieve relatively precise control of the spatial distribution of brain stimulation. However, even with this approach, the induced current intensity is not uniformly distributed throughout rIPS (see *Figure 8*). IPS is both narrow and relatively long. The current-flow estimate of the stimulation we delivered suggests the strongest stimulation arises in the most posterior regions of IPS (IPS1 and IPS2[4]); however, auditory spatial attention relies primarily on higher IPS regions (*Michalka et al., 2016*). Additional studies should be undertaken that more precisely target areas engaged by auditory spatial attention, for instance, by designing HD-tACS protocols using current flow modeling based on each individual subject's anatomy. New methods for improving the spatial precision of alternating current stimulation, such as temporal interference or interferential stimulation, could also lead to more robust stimulation of cortical structures, particularly at depth (*Huang and Parra, 2019*; *Rampersad et al., 2019*).

Some past studies suggest that stimulation can cause neural entrainment, synchronizing intrinsic oscillations to the external input (e.g., see *Helfrich et al., 2014*). External stimulation also has been

shown to modulate the spontaneous activity of neurons by shifting resting cell membrane potentials (*Purpura and Mcmurtry, 1965*; *Nitsche et al., 2003*), directly influencing endogenous brain oscillations (*Bergmann et al., 2009*; *Zaehle et al., 2010*). Unfortunately, our stimulation hardware did not allow us precise control over stimulation timing. We therefore adopted a block stimulation design rather than applying stimulation based on real-time neural measures. Thus, while we demonstrate that alpha stimulation directly alters spatial auditory attention, further experiments are needed to clarify how external tACS interacts with intrinsic oscillations. For instance, with more flexible hardware, one could explore whether changing the phase of tACS relative to ongoing, intrinsic oscillations alters how stimulation affects performance. Measuring simultaneous electroencephalography (EEG) during or even immediately after stimulation would also provide more direct insights into how stimulation influences neural responses. Of course, such an approach would require removal of electrical artifacts caused by stimulation, which is a significant technical challenge (e.g., see *Neuling et al., 2017*; *Witkowski et al., 2016*; *Noury and Siegel, 2017*). Future work is needed to map out the physiological foundations of tACS and the relationship between stimulation effectiveness, effect duration, and tACS parameters such as current intensity and frequency.

There is no consensus as yet on whether or not tACS induces neural entrainment that persists beyond the period of active stimulation (*Antal and Paulus, 2013*). Some human studies report tACS after-effects that persist as long as 40 min (*Helfrich et al., 2014*; *Reinhart, 2017*; *Wischnewski and Schutter, 2017*). In vivo animal studies show that neural entrainment ceases as soon as stimulation stops (*Ozen et al., 2010*; *Deans et al., 2007*); yet, animal studies also generally use briefer stimulation periods, which may reduce the duration of any after-effects (*Strüber et al., 2015*). In our study, the effects of HD-tACS did not persist into post-stimulation testing (*Figure 6*). It may be that more intense stimulation would elicit an after-effect (*Moliadze et al., 2012*). Alternatively, alpha stimulation effects may be more prominent and after-effects more long-lasting when the exact frequency of stimulation is matched to the individual subject's natural peak alpha frequency (*Reinhart and Nguyen, 2019*).

While we were able to degrade attention to sounds contralateral to the stimulation site, we did not find enhancement of ipsilateral attention; however, we did not match our stimulation to individual participants' alpha frequency. It may be that when the driving frequency of tACS stimulation matches the intrinsic oscillation frequency of a particular neural assembly (such as the assembly responsible for intrinsic parietal alpha oscillations), stimulation will be more effective at entraining relevant neural responses. If so, matching the tACS stimulation to an individual's measured parietal alpha oscillations may preferentially influence the relevant parietal neural population involved in steering spatial attention. Closed-loop methods that individually match the frequency of HD-tACS to endogenous neural oscillations in real-time thus may provide more robust changes in brain function. Such approaches could open up an entirely new realm of treatment options for cognitive brain disorders.

The present findings add to a growing body of neuromodulation research addressing the importance of rhythmic neural information in health and disease. Abnormalities in parietal alpha and its distribution across cerebral hemispheres have been documented in numerous disorders (e.g., Alzheimer's disease [*Koelewijn et al., 2017*], depression [*Bruder et al., 1997*] and ADHD [*Hale et al., 2009*]). Studies like ours, which directly manipulate oscillatory parietal activity, lay critical groundwork for development of interventions to alleviate problems due to atypical cortical mechanisms in a range of neurological and neuropsychiatric disorders.

## Materials and methods

### Subjects

Thirty-eight volunteers gave written consent to procedures approved by the Boston University Institutional Review Board. The subjects were paid for their participation. Twenty volunteers aged between 18–24 years (mean 21.15 year., standard deviation 3.01 year.; 13 females, seven males) participated in the main experiment. Eighteen volunteers aged between 18–24 years (mean 22.11 year., standard deviation 2.4 year.; 15 females, three males) participated in the control experiment. All participants had clinically normal audiometric thresholds in both ears for frequencies from 250 Hz to 8 k Hz (thresholds less than 20 dB HL).

We conducted a pilot experiment with six subjects and estimated effect size for the 'attend left, continuous, spatial attention' trials during alpha stimulation of right parietal cortex. Given that we also planned on testing 'attend right, continuous, spatial attention' trials, we estimated that we needed 14 subjects to achieve a power of 0.903 (correcting the family-wise error rate for multiple comparisons). Because we anticipated some attrition, we recruited 20 subjects in the main experiment and 18 in the control experiment, all of whom completed the experiments.

## Task and stimuli

Subjects performed a selective auditory attention task, diagrammed in *Figure 2A*. At the beginning of each trial, subjects fixated their gaze on a dot presented at the center of a computer screen. A visual cue starting 0.5 s later (0.4 s duration) instructed subjects which of two competing speech streams to attend, based on either *spatial* or *nonspatial* features. In *spatial* trials, the cue was either a left or right arrow, indicating the location of the target speech stream. In *nonspatial* trials, the cue was either an up or down arrow, indicating the gender of the target stream talker (female or male talker). The sound stimuli began 1.2 s after the visual cue disappeared and lasted 2.3 s. Listeners had one second after this to enter their response on the keyboard, at which point a colored circle appeared around the fixation dot for 0.2 s to indicate if the response was correct (blue) or incorrect (red). A small cash reward was given to subjects for each trial correctly answered within the time limit to help ensure subjects remained attention throughout the session.

In each trial, sound stimuli comprised two competing speech streams: a target and a distractor. Both target and distractor streams were five syllables long. The syllables were chosen from the same set of voiced-stop-consonant-vowel utterances (/ba/,/da/, and/ga/; each 388 ms in duration) recorded by one female talker and one male talker (F0 roughly 189 Hz and 125 Hz, respectively, as estimated by Praat software). Each syllable was spatialized to be perceived as either 90 degrees to the left or 90 degrees to the right by convolving raw recordings with manikin head-related transfer functions (*Gardner and Martin, 1995*). Both the target stream and the distractor stream were isochronous, with an inter-syllable interval of 433 ms. However, the two streams were temporally interdigitated: the distractor stream always began first, 180 ms before the target stream.

On each trial, the listener's goal was to count the number of/ga/ syllables in the target stream, which was either defined by its location in *spatial* trials (left or right, chosen randomly on each trial with equal likelihood) or its talker in *nonspatial* trials (male or female, chosen randomly on each trial with equal likelihood). The task-irrelevant feature (talker in *spatial* trials; location in *nonspatial* trials) was either consistent throughout the stream (*continuous* trials) or changed from syllable to syllable within both the target stream and the distractor stream (*switching* trials; see *Figure 2B*). Each trial was pseudo-randomly chosen to be either *spatial* or *nonspatial* and either *continuous* or *switching*. Therefore, to perform the task, listeners had to focus attention on the task-relevant dimension for that trial and to try to ignore the task-irrelevant dimension. The five syllables making up each stream were randomly chosen with replacement, subject to the further constraint that no syllable was the same as the syllable just prior to it or just after it (which were always in the competing stream).

Stimuli were presented via earphones (ER-2, Etymotic Research, Inc) in a double-walled Eckel sound-treated booth at Boston University. All sound stimuli were presented at a sound pressure level of approximately 75 dB.

## Experimental procedures

We conducted two experiments, differing in the form of HD-tACS stimulation that was applied to right parietal cortex. In the main experiment, HD-tACS stimulation was at a frequency in the middle of the alpha range (10 Hz), while in the control experiment, it was in the theta frequency range (6 Hz). Other than this detail, the two experiments were procedurally identical.

In each experiment, each subject performed two experimental sessions on two different days: a 1.5 mA HD-tACS Stimulation session and a Sham control session. The order of the Stimulation and Sham sessions was counterbalanced across subjects. In the main experiment, the two testing days were separated by 1–14 days (mean of 2.45 days; standard deviation of 3.12 days). In the control experiment, the two testing days were separated by 1–16 days (mean of 4.61 days; standard deviation of 5.04 days). Subjects were blinded to the stimulation order.

Each session consisted of a brief training period, followed by three formal data collection blocks of 208 trials, each of which lasted approximately 20 min (see *Figure 2C*). Training at the start of each session consisted of two mini-blocks to ensure that subjects understood the task. In the first training mini-block, subjects performed practice runs of 40 trials in which they counted the number of/ga/ syllables appearing in a 5-syllable target stream presented in quiet until they reached an accuracy of 80%. In the second training mini-block, subjects performed practice runs of 48 trials identical to those in the formal experimental attention task until their accuracy reached 50% (chance level on this task is 17%). Subjects were allowed to repeat these training runs until they reached this criteria. No subjects failed to successfully meet the criteria. The majority of the subjects reached this criteria on each of the training tasks after only 1–2 runs; however, one subject required 8 runs of the with-distractor training. The results of this subject, however, did not stand out in any way from the results of the other subjects.

Following training in each session, subjects conducted three 20 min long blocks of 208 trials (see *Figure 8B*). The first block served as a baseline control, with no neural stimulation. In the second block, subjects received either HD-tACS stimulation or sham stimulation for 20 min. No stimulation was performed in the final, third block, which allowed us to evaluate whether there were any aftereffects of stimulation. Because alpha frequency stimulation has been reported to be more effective in darkness (*Kanai et al., 2008*), all subjects but one performed the experiment in the dark; the exception reported suffering from claustrophobia and performed the task in typical lighting.

The trial order within each 208-trial block was pseudo-random, with each trial type equally likely. Each block contained exactly 104 trials of *spatial* attention trials (52 attend left and 52 attend right) and 104 trials of *nonspatial* attention (52 attend female and 52 attend male). Additionally, half of the trials were *continuous* and the other half *switching* (e.g., each of the three blocks in an experimental session contained exactly 26 trials of *attend left, continuous* trials). To avoid fatigue, subjects were given a 10 s rest period after every 48 trials within each block.

## High definition transcranial alternating current stimulation (HD-tACS)

HD-tACS was administered using the Soterix M × N-9 High Definition-Transcranial Electrical Current (HD-tES) Stimulator (Model 9002A, Soterix Medical, New York, NY). To target right IPS, an electrode montage was created based on a current-flow model generated by the Soterix HD-Explore software (version 4.1, Soterix Medical, New York, NY). Stimulation electrodes were placed in HD Electrode holders (Soterix Medical, New York, NY) and embedded in a 64-channel EEG cap. The electrode holders were filled with gel to ensure impedance for each electrode did not exceed 50 k ohms prior to stimulation and remained below 5 k ohms during stimulation (*Thair et al., 2017*). The major stimulating electrode was placed at P2 with a stimulation intensity of 1.5mA, and four return electrodes were placed at CP2 (−0.6mA), P4 (−0.225mA), Pz (−0.075mA), and PO4 (−0.6mA). Both the HD-tACS stimulation and sham sessions used the same electrode montage. *Figure 8A* depicts the electrode placement of the montage and simulated current-flow model.

Both Stimulation and Sham sessions delivered a bipolar sinusoidal waveform at 10 Hz (*Figure 8B*). Despite the fact that there are individual differences in peak frequencies of oscillation activity, such as in alpha (*Haegens et al., 2014*), we chose to stimulate at the same frequency for all subjects. When targeting parietal alpha oscillations in the main experiment, we chose a 10 Hz stimulation rate, which is close to the peak reported for most subjects (in the 10–11 Hz range). The control experiment used a 6 Hz rate, which is the median peak theta frequency (*Jacobs, 2014*).

The total current delivered was 1.5 mA at maximum. While the most effective intensity and duration for HD-tACS or traditional tACS stimulation is not known, previous tACS studies have commonly used 1.5–2 mA (*Helfrich et al., 2014*; *Reinhart and Nguyen, 2019*; *Klaus and Schutter, 2018*). With very few studies to reference on the effectiveness of HD-tACS intensity, we arbitrarily chose a relatively conservative and widely used stimulation intensity of 1.5 mA to mitigate any adverse effect of stimulation (*Matsumoto and Ugawa, 2017*).

In the HD-tACS session, stimulation ramped up to 1.5 mA over 30 s at the beginning of the 20 min stimulation block, and ramped down over 30 s at the end, yielding 19 min of continuous 1.5 mA stimulation during the middle of the block. During the middle block of the Sham session, stimulation ramped up to 1.5 mA over 30 s and then immediately ramped down to 0 mA in the following 30 s; in the final minute of the block, stimulation ramped up and then down. During the 40 s at the

beginning of stimulation in both sessions, subjects were verbally checked to ensure they were familiar with the stimulation-induced sensation and that they were comfortable proceeding with the experiment.

In our pilot experiments, we asked subjects verbally whether they could differentiate sham and stimulation sessions. None could. This confirmed the experience we have from our past HD-tACS studies using comparable stimulation parameters (including intensity, duration, sink-source electrode configuration, low frequencies), where we systematically probed the subjects' ability to differentiate sham from stimulation; in no case were responses significantly different from random guessing (*Reinhart and Nguyen, 2019*; *Reinhart, 2017*; *Nguyen et al., 2018*).

## Statistical analysis

To test our hypotheses, we calculated the percentage of correct responses for each attention condition (*spatial* attention: attend left vs. right; *nonspatial* attention: attend female vs. male). We then baseline corrected for each attention condition in the during-stimulation block and post-stimulation block by subtracting the accuracy of the corresponding trial type during the initial baseline block. For the pairwise comparisons of accuracy for the two a priori planned statistical tests (i.e., leftward spatial attention in continuous trials should be hurt by alpha stimulation and rightward spatial attention in continuous trials might be enhanced by alpha stimulation), single-tailed Wilcoxon signed rank tests were performed (significance for $p<0.05$) and Bonferroni corrected for multiple corrections. For the many control conditions, where we expected no effects, we did post hoc Wilcoxon signed rank tests, without correction for multiple comparisons (which is conservative when we expected no effect, as correction would make it more likely that we dismissed marginal effects).

## Acknowledgements

We thank Charlotte Xiyou Wang, Ashvini Melkote and Ray Lefco for help with data collection. This work was supported by National Institutes of Health Grant R01 MH-114877 to RMGR, National Institutes of Health Grant R01 DC015988 to BGSC, and Office of Naval Research Grant N00014-18-1-2069 to BGSC.

## Additional information

### Competing interests

Barbara G Shinn-Cunningham: Senior editor, *eLife*. The other authors declare that no competing interests exist.

### Funding

| Funder | Grant reference number | Author |
| --- | --- | --- |
| National Institutes of Health | R01 DC015988 | Barbara G Shinn-Cunningham |
| Office of Naval Research | N000141812069 | Barbara G Shinn-Cunningham |
| National Institutes of Health | R01 MH-114877 | Robert MG Reinhart |

The funders had no role in study design, data collection and interpretation, or the decision to submit the work for publication.

### Author contributions

Yuqi Deng, Conceptualization, Data curation, Software, Formal analysis, Investigation, Visualization, Methodology; Robert MG Reinhart, Conceptualization, Resources, Formal analysis, Supervision, Methodology; Inyong Choi, Software, Formal analysis, Methodology; Barbara G Shinn-Cunningham, Conceptualization, Resources, Supervision, Funding acquisition, Visualization, Methodology, Project administration

## Author ORCIDs
Yuqi Deng (iD) https://orcid.org/0000-0001-7278-0841
Robert MG Reinhart (iD) https://orcid.org/0000-0003-2156-4633
Inyong Choi (iD) https://orcid.org/0000-0002-6663-9152
Barbara G Shinn-Cunningham (iD) https://orcid.org/0000-0002-5096-5914

## Ethics
Human subjects: All subjects gave informed consent, as approved by the Boston University Charles River Campus IRB, under protocol 3597E.

## Decision letter and Author response
Decision letter https://doi.org/10.7554/eLife.51184.sa1
Author response https://doi.org/10.7554/eLife.51184.sa2

## Additional files

### Supplementary files
• Transparent reporting form

### Data availability
Data are available from Dryad at https://doi.org/10.5061/dryad.c031nv7.

The following dataset was generated:

| Author(s) | Year | Dataset title | Dataset URL | Database and Identifier |
|---|---|---|---|---|
| Deng Y, Shinn-Cunningham B, Choi I, Reinhart R | 2019 | Data from: Causal links between parietal alpha activity and spatial auditory attention | https://doi.org/10.5061/dryad.c031nv7 | Dryad Digital Repository, 10.5061/dryad.c031nv7 |

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
