## [Decision Letter]

**Acceptance summary:**

Lateralized alpha-band neural oscillation has been widely proposed to mediate spatial attention in both visual and auditory tasks. However, there is little causal evidence, particularly in auditory spatial attention. This work used a HD-tACS approach aiming to achieve focused, frequency-specific stimulation of human parietal cortex, when subjects perform a series of auditory attention task. The study is well motivated and clearly proposes their prior hypothesis based on previous findings, as summarized and illustrated in Introduction. Specifically, the authors expect to see a particular interference effect, that is, alpha-band (instead of other frequencies) stimulation in the right parietal region would disrupt performance when subject attend to the left side. They further hypothesized an absence of the disruption effect in switching task when top-down spatial attention is disrupted.

Their HD-tACS results demonstrate the specific effect, thus constituting novel evidence supporting the direct, causal relationship between neural oscillations in human parietal cortex and control of auditory spatial attention. The effect is specific not only to task demands but also to both the frequency of stimulation and the direction of attention. Linking to previous findings in visual spatial attention, this work provides important evidence for a unified theoretical system of visual and auditory spatial attention.

**Decision letter after peer review:**

Thank you for submitting your article "Causal links between parietal alpha activity and spatial auditory attention" for consideration by *eLife*. Your article has been reviewed by three peer reviewers, and the evaluation has been overseen by a Reviewing Editor and Floris de Lange as the Senior Editor. The following individual involved in review of your submission has agreed to reveal their identity: Adrian K C Lee (Reviewer #1).

The reviewers have discussed the reviews with one another and the Reviewing Editor has drafted this decision to help you prepare a revised submission.

Summary:

This work used HD-tACS approach to examine whether parietal alpha-band neuronal oscillations causally modulate auditory spatial attention. They demonstrate that the modulation effect specifically occurs under one condition, that is, alpha-band stimulation over right parietal region disrupts auditory spatial attention on the left side of space, when speakers come from one location continuously. All reviewers agree that this study is well motivated and addresses an important question in auditory neuroscience, aiming at testing one particular hypothesis. However, there are a few major issues brought up by the reviewers that need the authors to do additional analysis or discussion.

Essential revisions:

1) Statistical power concerns. The authors have examined the effects under many conditions, including stimulation frequencies (alpha vs. theta), stimulation hemisphere (left vs. right), attended features (speaker vs. space), and auditory contexts (continuous vs. switching), and found significant effect for only one condition. This, on the one hand, nicely supports the specific effect, but on the other hand, also introduces worries about the statistical power of the effect. The authors should explicitly specify whether the reported p-values are multiple-comparison corrected or uncorrected and what type of correction has been used (e.g., Bonferroni, FDR, or other methods). For example, it is unclear whether the p-value is corrected for the 2 conditions shown in Figure 5 and in Figure 7, and whether the results shown in the second paragraph of the subsection “Α HD-tACS of rIPS disrupts auditory spatial attention for leftward targets” include a correction for rightward attention condition. Moreover, in Table 1, Padj is shown for the first 2 conditions while P is shown for other conditions. The authors should make the p name consistent and explicitly specify the meaning of Padj. If the authors have a quite clear hypothesis about one specific effect, for example, for leftward but not rightward attention, they should justify their hypothesis in more details in Introduction.

2) What is the potential neural mechanisms? For example, why is alpha-rate tACS expected to enhance rather than suppressing alpha oscillations? How does the tACS alpha stimulation interacts with the intrinsic alpha-band neural oscillations?

3) The paper does not reveal effects for the switching condition which indeed requires strong spatial attention. This is surprising and needs more explanation. Moreover, subjects might depend on the speaker's voice instead of genuinely using spatial attention in the continuous attention condition to perform the task. It is possible that if the authors implement an experiment that genuinely and specifically requires subjects to deploy spatial attention, the spatial attention effects might be observed in the right hemifield. The authors should add discussions on the limitations of the current design (i.e., a task that can only be solved by deploying true spatial attention and that subjects can actually do).

---

## [Author Response]

Essential revisions:1) Statistical power concerns. The authors have examined the effects under many conditions, including stimulation frequencies (alpha vs. theta), stimulation hemisphere (left vs. right), attended features (speaker vs. space), and auditory contexts (continuous vs. switching), and found significant effect for only one condition. This, on the one hand, nicely supports the specific effect, but on the other hand, also introduces worries about the statistical power of the effect. The authors should explicitly specify whether the reported p-values are multiple-comparison corrected or uncorrected and what type of correction has been used (e.g., Bonferroni, FDR, or other methods). For example, it is unclear whether the p-value is corrected for the 2 conditions shown in Figure 5 and in Figure 7, and whether the results shown in the second paragraph of the subsection “Alpha HD-tACS of rIPS disrupts auditory spatial attention for leftward targets” include a correction for rightward attention condition. Moreover, in Table 1, Padj is shown for the first 2 conditions while P is shown for other conditions. The authors should make the p name consistent and explicitly specify the meaning of Padj. If the authors have a quite clear hypothesis about one specific effect, for example, for leftward but not rightward attention, they should justify their hypothesis in more details in Introduction.

The Introduction now better explains our hypotheses. The Materials and methods section now explains that our a priori tests were corrected for multiple comparisons (Figure 5), but that we (conservatively) did *not* do a multiple-comparisons correction for the control conditions, where we expected no effect. (That is, such corrections for the myriad control conditions actually would make it *more* likely for us to overlook marginal effects; since our hypotheses for these conditions was a lack of an effect, correcting for multiple comparisons is *less* conservative than leaving them uncorrected.) To further emphasize that only during alpha stimulation did we expect any effects, we have renamed what was “Experiment 2” (using theta stimulation) “the control experiment,” and refer to what was “Experiment 1” as “the main experiment” throughout the manuscript.

2) What is the potential neural mechanisms? For example, why is alpha-rate tACS expected to enhance rather than suppressing alpha oscillations? How does the tACS alpha stimulation interacts with the intrinsic alpha-band neural oscillations?

In conditions where top-down attentional control does not lead to strong alpha, we expect stimulation to produce neuronal entrainment at the population level (e.g., see Helfrich et al., 2014), driving strong alpha (i.e., when listeners are asked to attend to the left, and we stimulate in right parietal cortex). That is, we propose that stimulation capitalizes on the self-sustained and dynamic property inherent to oscillatory neurons by providing a periodic force to which the natural oscillation becomes synchronized.

In the case where top-down attention itself causes strong alpha in the parietal cortex that is stimulated, we are less sure about how stimulation will affect neural responses. In particular, past work suggests evidence that stimulation is less effective when intrinsic oscillatory power is high (as it should be in rIPS during rightward attention). Moreover, there may be interactions between intrinsic alpha band oscillations and stimulation that depend on their relative phase. This initial study does not give us any direct evidence for how the stimulation interacts with intrinsic oscillations; any arguments we might make would be pure speculation and go beyond the scope of the study. Specifically, we did not measure EEG during the experiment — although that is something we hope to do in the future.

The Discussion subsection “Caveats and Future Work” now addresses all of these issues. For instance, we explain that we could not control stimulation timing to directly test how external stimulation interacts with intrinsic activity. We thus cannot determine how tACS alpha stimulation interacts with any intrinsic alpha-band neural oscillations. We note that future studies using real-time control of stimulation, including matching each individual’s measured alpha activity, and new methods for achieving more spatially precise stimulation, should be undertaken to address these important, open questions.

3) The paper does not reveal effects for the switching condition which indeed requires strong spatial attention. This is surprising and needs more explanation.

To us, this result was not at all surprising, but rather exactly what we expected. As we discuss in the Introduction, we believe, based on past results by Mehraei, that when a talker switches direction, it disrupts top-down spatial attention (exhibited by both a disruption of alpha lateralization at the moment of the switch, a symptom of the spatial-attention failure, and an increase in response errors at that moment). Our behavioral results in the initial, pre-stimulation trials support the idea that *topdown spatial attention* is disrupted in the switching condition. That is, listeners might *require* strong spatial attention to do well in switching trials, but they cannot achieve it and make behavioral errors. They *do not* maintain strong top-down spatial attention in this condition— and parietal alpha lateralization reflects this— despite their best intentions.

Our hypothesis, borne out by our behavioral results in the pre-stimulation trials, is that switches in talker location disrupt top-down attention automatically (the talker switches cause a “bottom up” interruption of volitional control of spatial attention). This “automatic override” of top-down spatial attention in the switching condition operates not only in the “natural” listening state, but during external parietal stimulation. Specifically, talker switching disrupts parietal-cortex mediated topdown spatial control through some automatic, bottom-up pathway. Because the bottom-up disruption of spatial attention bypasses top-down spatial attentional control, parietal stimulation is irrelevant in that case. We have tried to explain this hypothesis more clearly both in the Introduction and in the Discussion.

Moreover, subjects might depend on the speaker's voice instead of genuinely using spatial attention in the continuous attention condition to perform the task.

We now state in the Introduction that

“Subjects were instructed and explicitly aware that in both the spatial and talker attention blocks, each trial was equally (and unpredictably) likely to be a switching trial or a continuous trial. They had no way of predicting which trials would be switching and which continuous; therefore, there was no way for them to have adopted different listening strategies for the continuous and the switching trials within a block. Furthermore, they were very aware (and explicitly instructed) that attending to voice was not a reliable strategy in the attend-location block.”

We agree that this was not clear in the original manuscript. This revised version of the manuscript makes these points and this logic explicit.

Consistent with the assertion that listener used spatial attention in attend-location blocks, we find an effect of parietal stimulation during these blocks. This result supports the idea that listeners must have been using the spatial-parietal attention network (i.e., genuinely using spatial attention) in the continuous as well as the switching trials of the spatial attention condition.

It is possible that if the authors implement an experiment that genuinely and specifically requires subjects to deploy spatial attention, the spatial attention effects might be observed in the right hemifield. The authors should add discussions on the limitations of the current design (i.e., a task that can only be solved by deploying true spatial attention and that subjects can actually do).

The “Caveats and Future Work” subsection of the Discussion now includes an expanded discussion of the reasons why there may not be observable effects of stimulation when attention is directed rightward. However, we do not believe it is related to the form of attention that subjects deployed. Instead, it could be due to many other limitations: the fact that intrinsic alpha power is likely already high in that condition, because there are limitations in the specificity and precision of the stimulation we can achieve, because of limited statistical power, etc.

We are quite confident that spatial attention is required, and utilized, by listeners in all “attend location” trials of the current design. As described above, location was the only consistent, reliable feature to allow listeners to perform the task in attend location blocks. They could not rely on voice, as they could not predict which trials were switching and which continuous. Again, we now describe the design of the study more explicitly in the Introduction. If the clarifications and additional caveats described in this revised version of the manuscript do not adequately clear up the confusion, we would be happy to make further changes to the manuscript.